# Distribution Structure Learning Loss (DSLL) Based on Deep Metric Learning for Image Retrieval

**DOI:** 10.3390/e21111121

**Published:** 2019-11-15

**Authors:** Lili Fan, Hongwei Zhao, Haoyu Zhao, Pingping Liu, Huangshui Hu

**Affiliations:** 1College of Computer Science and Technology, Jilin University, Changchun 130012, China; llfan18@mails.jlu.edu.cn (L.F.); zhaohw@jlu.edu.cn (H.Z.); 2Key Laboratory of Symbolic Computation and Knowledge Engineering of Ministry of Education, Jilin University, Changchun 130012, China; 3Editorial Department of Journal (Engineering and Technology Edition), Jilin University, Changchun 130012, China; zhaohaoyu@jlu.edu.cn; 4School of Computer Science and Engineering, Changchun University of Technology, Changchun 130012, China

**Keywords:** deep metric learning, entropy weight, fine-tune network, image retrieval, structural preservation, structural ranking consistency

## Abstract

The massive number of images demands highly efficient image retrieval tools. Deep distance metric learning (DDML) is proposed to learn image similarity metrics in an end-to-end manner based on the convolution neural network, which has achieved encouraging results. The loss function is crucial in DDML frameworks. However, we found limitations to this model. When learning the similarity of positive and negative examples, the current methods aim to pull positive pairs as close as possible and separate negative pairs into equal distances in the embedding space. Consequently, the data distribution might be omitted. In this work, we focus on the distribution structure learning loss (DSLL) algorithm that aims to preserve the geometric information of images. To achieve this, we firstly propose a metric distance learning for highly matching figures to preserve the similarity structure inside it. Second, we introduce an entropy weight-based structural distribution to set the weight of the representative negative samples. Third, we incorporate their weights into the process of learning to rank. So, the negative samples can preserve the consistency of their structural distribution. Generally, we display comprehensive experimental results drawing on three popular landmark building datasets and demonstrate that our method achieves state-of-the-art performance.

## 1. Introduction

Along with the popularity of the Internet and smart devices, the number of pictures stored on the network has exploded. Therefore, image retrieval has become a common concern. One big challenge in image retrieval is to detect robust and discriminative features from numerous images. Traditional methods depend on handcrafted features, which include global features such as spectral (color), texture, and shape features, as well as the aggregated features like bag of words (BoW) [1], vector of locally aggregated descriptors (VLAD) [2], and Fisher vector (FV) [3]. Such design is time-consuming and needs substantial professional knowledge.

The advancement of deep learning has pushed forward the development of content-based image retrieval [4]. Deep convolutional neural network (CNN) features become greatly abstractive and contain semantic information at high-level, which have outperformed traditional handcrafted features in image retrieval [5,6]. Furthermore, the deep features are learned automatically from data and there is no need to design the features by labor work, which makes deep learning techniques extremely valuable in large-scale image retrieval.

As emerging techniques, deep metric learning (DML) combines deep learning and metric learning [7], and includes batch metric learning approaches [8] and large scale online metric learning approaches [9]. DML deploys its discriminative power of deep neural networks into an embedding metric space where the semantic similarity between images will be directly measured by simple metrics, such as Euclidean distance. Deep metric learning is proven to be effective in fields like transfer learning [10], face recognition [7,11], person re-identification [12], and natural image retrieval [13].

The main purpose of metric learning is to aim to learn a new metric to reduce the distances between samples of the same class and increase the distances between samples of different classes [14]. Some kind of hard example mining and loss function can also be used [15]. The loss function is crucial in successful DML frameworks and various kinds of loss functions have been put forward in past works. Contrast loss [16,17] discards negative pairs whose similarities are smaller than a given threshold by capturing the distance relationship between pairs of data points. Numerous studies have focused on triplet-based losses [7,18,19], which is composed of an anchor point, here referring to a positive (similar) data point and negative (dissimilar) data point. Triplet loss aims to learn a distance metric through which the above anchor point is closer to a positive point than the negative one by a margin. Generally, the triplet loss considers the relationship between positive and negative pairs so it outperforms the contrastive loss [7,10]. Inspired by this, many updated studies have begun to consider richer structured information between multiple data points and have achieved satisfying performance on many applications [7,10,20,21,22].

However, the current state-of-the-art DML approaches still have certain limitations and need improvement. Firstly, for a known query image, only a small amount of data is combined to train the network and calculate the loss, and pull the samples with low similarity to the same distance from the query image, ignoring some useful examples and structural distribution information [23]. We select five negative samples with low similarity and propose entropy weight based on structural distribution. By assigning weights to negative samples according to the distribution of samples around negative samples, the structure distribution ranking learning is carried out, which can achieve satisfactory retrieval accuracy.

Second, the previous structured losses do not consider intra-class distribution [23]. The purpose of all algorithms [2,10,20,21] is to bring the samples with high similarity as close as possible to the query sample. Consequently, these approaches try to shrink samples of the high sample into one point in the feature space. This method can easily discard the structure of the sample. In order to solve this problem, we recommend learning a hypersphere for each query sample in the model. Specifically, for a positive sample, we simply force the distance between the positive sample and the query image to be less than the threshold, rather than pulling the positive sample more and more compact. In this case, the DSLL can help preserve the similarity structure of each sample and the ordinal relationship of the data as much as possible. To sum up, the major significance and innovations are clarified as follows.
We propose a novel and effective metric learning method. To the best of our knowledge, it introduces the concept of structural preservation and structural consistency for the first time. Structural preservation deals with the intraclass distribution of positive samples, while structural consistency accounts for the structural distribution information of negative samples.To solve the problem of shrinking each feature vector of positive samples into the same point, we set a hypersphere learning for every class. The method uses the hardest positive sample to train the network by adjusting its distance from the query sample. This effectively preserves the structure of the positive sample.We add the entropy weight based on structural distribution to the loss function. The algorithm considers the distribution characteristics of the negative samples, captures the feature structure, addresses the relative similarity of each sample, which has been ignored in past works, and calculates the loss by using weight to rank the negative samples. The proposed method deals well with sample similarity and structural distribution of each sample.We present comprehensive experiments on the popular Oxford buildings [1], Paris [24], and Holiday [25] datasets and the results demonstrate that our method achieves state-of-the-art performance.

The rest of this paper is organized as follows. We firstly review some deep metric learning methods and briefly introduce its existing applications in image retrieval in Section 2. Section 3 describes the algorithm framework and its detail. Then, Section 4 discusses the major significance and evaluates our newly proposed network architecture by a series of targeted experiments, after which future works of this paper are drawn in the final section.

## 2. Related Work

### 2.1. Image Retrieval 

Search accuracy and search speed play a key role in image retrieval in which search accuracy relies on image characteristics, while search speed is largely determined by indexing. In the early stages of image retrieval, image features were acquired by local patch descriptors (e.g., SIFT [26]) and SURF [27]. The image-level signatures are obtained by BoW, as the aggregated [28] local features [24,29].

This feature of the inverted table is further used as a valid index structure. Then, FV and average pooling (SPoC) [30] are used to aggregate the local features of the image to form an image global feature [30]. For feature aggregation, Tolias et al. [31] proposed maximum activations of convolutions (MAC) descriptors extracted from many multi-scale overlapping regions of the last convolutional feature map. The region descriptor is processed by L2 normalized, then through principal component analysis (PCA) + whitened [32], and finally normalized by L2 and aggregated into a global signature called regional maximum activations of convolutions (RMAC) [33].

The above methods all use fixed pre-trained CNNs. But these CNNs are trained to classify images, and a case in point is 1000 classes of ImageNet. This does not differentiate the aggregation method, and more inclined to sub-optimally perform in image retrieval. In order to solve this question, Radenovic et al. [34], proposed to add the MAC layer to the last convolutional layer of VGG [35] or ResNet [36], then used the Siamese structure to train the network [34]. In [37], a trainable generalized mean (GEM) pool layer replaces the MAC layer. This greatly improves retrieval accuracy. The recent NetVLAD consists of a standard CNN and a subsequent VLAD layer vector that aggregates the final convolutional features into fixed dimensional signatures and whose parameters can be trained by backpropagation [38]. To decrease the co-occurrences of certain weights and handle the issue of over-counting, J´egou and Chum [32] considered it was essential to whiten the data representation. A generative model based on an unsupervised approach through PCA conducted on an independent dataset [24,39] helps to learn whitening in the above process. Also, Radenović proposed a discriminative manner in whitening transform. This is achieved by using the same procedure of acquisition from the training data based on a 3D model. In image retrieval, query extensions based on CNN global descriptors [13,37] can be used to improve search accuracy.

### 2.2. Deep Metric Learning

One purpose of metric learning aims to learn a nonlinear projection function that converts an image from pixel level to discriminant space, where samples from the same category or high similarity will be collected together and samples from different categories or samples with low similarity will be pushed away. Recent research on depth metric learning in the visual tasks of face recognition [40], pedestrian recognition [41], image set classification [42], fine-grained retrieval [43], image retrieval [44], target tracking [45], multimedia retrieval [46], and visual search [47,48] have played a catalytic role. We will summarize the latest in-depth metrics learning methods in the following sections.

#### 2.2.1. Contrastive Loss

By minimizing the loss function, the gap (distance) between positive samples is gradually narrowed while the gap (distance) between negative samples is gradually widened.
(1)$$L_{c}=Yd_{I_{a},I_{b}}^{2}+\left(1-Y\right){\left(\alpha -d_{I_{a},I_{b}}\right)}_{+}^{2}.$$

This loss overruns pairs of samples. Let $I_{a}{,\;I}_{b}\in I$ be a pair of input images shown to the datasets. Through the pre-feedback of the network, we can get their normalized feature vector $f_{I_{a}}$ and $f_{I_{b}}$, $d_{I_{a}{,I}_{b}}{=||f}_{I_{a}}\;-f_{I_{b}}||{}_{2}$ is the Euclidean distance of the two picture feature vectors, Y = 1 if $I_{a}$ and $I_{b}$ are considered similar, and Y = 0 if they are considered dissimilar. Here, ${\left(z\right)}_{+}$ means max (z, 0), α is a threshold parameter designed according to actual needs.

#### 2.2.2. Triplet Loss

In the feature space, the triplet loss narrows down the distance between the anchor point and the positive sample as close as possible and pulls away the distance between the anchor point and the positive sample as far as possible. Besides, the distance gaps between the anchor point and the negative sample are reflected within a threshold *m*: (2)$$L\left(X;f\right)=\frac{1}{\left|T\right|}\sum_{\left(a,b,c\right)\in T}{\left[d_{I_{a},I_{b}}^{2}+m-d_{I_{a},I_{c}}^{2}\right]}_{+},$$
where *T* is a triple set, *a*, *b* and *c* are the anchor points, the index of the positive and negative sample points, respectively, *f* is the embedded function, $d_{I_{a}{,I}_{b}}{\;=\;||f}_{I_{a}}\;-{\;f}_{I_{b}}||{}_{2}$ is the Euclidean distance of the eigenvectors of the two pictures $I_{a}$ and $I_{b}$, where ${\left(z\right)}_{+}$ denotes max (z, 0), m is a threshold parameter designed according to actual needs.

#### 2.2.3. N-Pair Loss

The N-pair loss is designed to take advantage of all sample pairs in mini-batch and learn more differentiated representations based on structural information between the data. Specifically, the sample consists of one positive sample and N−1 negative samples from N-1 different categories, i.e., one negative sample per category, and the loss function can be expressed as follows:(3)$$L=\frac{1}{N}\sum_{i=1}^{N}log\left(1+\sum_{i\neq j}exp\left(f{\left(X_{i}\right)}^{T}f\left(X_{j}^{+}\right)-f{\left(X_{i}\right)}^{T}f\left(X_{i}^{+}\right)\right)\right)+\frac{n}{m}\sum_{i=1}^{m}||f{\left(X_{i}\right)||}_{2},$$
where η is $L_{2}$ regularized embedding vector, N is the number of selected sample types, $X_{i}$ is the query image, $X_{i}^{+}$ is a sample to $X_{i}$, and $X_{j}^{+}$ is a sample with low similarity to $X_{i}$, where j≠i.

#### 2.2.4. Lifted Structured Loss

Lifted structured loss relies on the advantages of training batches of minibatch SGD training, uses random sampled image pairs or triples, constructs training batches to calculate the loss of each pairs or triplets, and the loss function is given as a log-sum-exp formulation:(4)$$L_{I_{i},I_{j}}=\log\left(\underset{\left(I_{i},I_{k}\in N\right)}{\sum }exp\left\{\alpha -D_{I_{i},I_{k}}\right\}+\underset{\left(I_{j},I_{l}\in N\right)}{\sum }exp\left\{\alpha -D_{I_{j},I_{l}}\right\}\right)+D_{I_{i},I_{j}}$$
(5)$$L=\frac{1}{2\left|P\right|}\underset{\left(I_{i},I_{j}\in P\right)}{\sum }\max{\left(0,L_{I_{i},I_{j}}\right)}^{2},$$
where $D_{I_{i}{,I}_{k}}=||f_{I_{i}}{\;-\;f}_{I_{k}}||{}_{2}$ is the Euclidean distance in the embedding space of samples $I_{i}{\;\mathrm{and}\;I}_{j}$, α is a threshold parameter designed according to actual needs. *P* is a set of similar sample pairs, and *N* is a set of dissimilar sample pairs.

#### 2.2.5. Proxy-NCA

By learning the anchor points, similarities, and dissimilarity proxy points, the triplet loss of the triplet loss in different spaces is optimized, and the ternary loss on the agent is the tight upper limit of the original loss. The Neighbourhood Components Analysis (NCA) loss attempts to use exponential weighting such that $I_{a}$ is closer to $I_{b}$ than to any element in the set *Z*. The loss can be expressed as follows:(6)$$L_{NCA}\left(I_{a},I_{b},Z\right)=-log\left(\frac{exp\left(-D\left(I_{a},I_{b}\right)\right)}{\sum_{I_{z}\in Z}exp\left(-D\left(I_{a},I_{z}\right)\right)}\right),$$
where $D_{I_{a}{,I}_{b}}\;=\;||f_{I_{a}}-{\;f}_{I_{b}}||{}_{2}$ is the Euclidean distance in the embedding space of samples $I_{a}$ and $I_{b}$, *Z* is the negative data set,$\;I_{b}$ and $I_{c}$ are the proxies of positive and negative points, the anchor is $I_{a}$. The NCA loss is much better than dynamic allocation because it uses static proxy allocation, which means that each class has a proxy. However, the agents in the static proxy assignment are learned during training and are displayed in the class vector of the fully connected layer classification. Therefore, proxy-NCA does not retain the scalability of DML because it needs to consider the number of classes.

The proposed DSLL is a structured loss based on structural consistency and structural distribution ranking consistency. It avoids two limitations of traditional methods by combining all meaningful data points and exploring inherent structured information. Figure 1 shows a comparison of our method with other different methods.

## 3. Proposed Method

The design of our DSLL algorithm solves two basic problems of deep metric learning for image retrieval: (1) we maintain the structure of the sample with high correlation with the query sample, and (2) we also preserve the structural similarity order consistency of the sample with low correlation with the query sample. The first novelty of our approach is to set the space between the positive sample and the given query picture so as to build up a more informative structure for learning features, thereby maintaining the structure of the positive sample feature vector. The entropy weight based on structural distribution is another major innovation put forward in our approach. Our basic idea is to calculate how the adjacent samples of each negative sample individual feature are distributed and use this distribution information to optimize the control of subsequent distance weighting operations. The entropy based on structural distribution refers to the relative relationship between the spatial distance distributions of the samples around the negative samples with low correlation. Hard negative samples provide higher discriminability between matching and non-matching pairs, and the sample structure information is more informative in the recognition, so they are given higher weight. In this section, we will elaborate on DSLL and its details proposed for image retrieval, which includes loss function and fine-tuned architecture of the two-branch network, and then we will also explain the principle of the loss function.

### 3.1. Network for Image Retrieval

#### 3.1.1. CNN Network Architecture

In order to obtain the eigenvectors of the image, we constructed a CNN network. As shown in Figure 2, we selected the convolutional layer of VGG (in the dashed box in Figure 2), which was completely convolved, and the network can extract features regardless of image size. The last pooling layer of the VGG convolutional layer was replaced by generalized-mean pooling (GeM). The feature map obtained by convolution is vectorized by the pooling operation, then the Lw whitening operation is performed, and finally L2-normalized. The VGG network is a fine-tuned network. In the following, the AlexNet [49] and ResNet101 [50] (pre-trained on ImageNet) network frameworks used in our experiments were formed by replacing the VGG part (in the dashed box) in Figure 2, where the network connection was the same as VGG. Finally, we removed the fully connected layer and replaced the last pooling operation with a GeM pooling operation.

#### 3.1.2. The Architecture of Network Training

As shown in Figure 3, the training part of the network is composed of multiple CNNs sharing the same weight. The CNN we used here is the network constructed in Figure 2. A positive sample is highly similar to a query image and belongs to the same category, while a negative sample is classified into a different category. The multiple identical networks adopted the corresponding feature vectors of input image multiples and output multiples. We combined the eigenvectors of the negative samples obtained with the obtained ordering, and then combined the query image with the eigenvectors of the positive samples to train the network.

#### 3.1.3. Network Evaluting Architecture

In the test section of this article, we build a network framework as shown in Figure 4. First, we adjusted the images in the query image and image database to different sizes through multi-scale processing, then put multiple input images to the CNN network, as the framework built in Figure 2, and form a single descriptor by combining the global descriptors from multiple scales. We calculate the size of the Euclidean distance between the image feature vector and the image feature vector in the image library, sort the image according to the similarity, and obtain the image with the top ranking. The result that does not match the query image is discarded. We sum the left results by using the original query and the renormalization of experience.

### 3.2. Distributed Structure Learning Loss

We aimed at learning the discriminant function $L_{m}$ and train the network so that a higher similarity is observed between positive sample pairs while a lower similarity between negative samples in the feature space.

Based on the similarity of pairwise in past works [26], our goal is to keep the positive sample closer to the predetermined structure and maintain the threshold boundary. Besides, we plan to separate the positive and negative samples with the threshold *β* and $e^{\frac{\alpha }{n}}\tau$, to achieve this, we take an architecture based on Siamese and train the network with two branches. The training input includes the image pair (q, i) and the label $y_{\left(q,\;i\right)}\;\in \;\left\{0,1\right\}$, if *q* and *i* is non-matching (label 0) or matching (label 1). Given an image $X_{i}$, our goal is to pull the positive sample closer to the predetermined structure retention threshold boundary *β*, pushing the negative sample to a different distance from the boundary τ according to the $e^{\frac{\alpha }{n}}$ value, where β and τ are the two boundary thresholds. We chose contrast loss to construct a loss function based on the consistency of distribution structure retention and structural similarity, and applied it to matching pairs and non-matching pairs, and the loss function is defined as
(7)$$L_{m}\left(X_{q},X_{i};f\right)=y_{\left(q,i\right)}{\left[d_{\left(q,i\right)}-\beta \right]}_{+}+\left(1-y_{\left(q,i\right)}\right)\cdot max\left\{0,e^{\frac{\alpha }{n}}\tau -d_{\left(q,i\right)}\right\}.$$

If *q* is similar to *i*, $y_{\left(q,i\right)}=1$, *q* and *i* are not similar, $y_{\left(q,\;i\right)}=0$, $d_{q,\;i}=||f\left(X_{q}\right)\;-\;f\left(X_{i}\right)||{\;}_{2}$ refers to the Euclidean distance between two points.

Selection of positive images: several sets of randomly selected images from positive image pairs were prepared during the training process. The annotated positive image pairs from the training dataset were treated as positive images inside the training sets. We used the method proposed in [37], the positive image pair was randomly selected from a set of images, rather than using a pool of images that are captured in a similar camera position. The image presented sufficient co-observed points with the query image, and do not show extreme scale changes. This positive imag
(8)$$m\left(q\right)=rnd\left\{i\in M\left(q\right):\frac{P\left(q\right)\cap P\left(i\right)}{P\left(q\right)}\geq t_{i},scale\left(i,q\right)\leq t_{s}\right\}.$$

In Equation (8), the scale changes between the two images are reflected from scale (*i, q*). The harder matching examples selected by this method are guaranteed to make sure the depiction of the same object. This method is conducive to increase more diversified viewpoints.

Selection of negative images: we calculated the Euclidean distance between the query image and the image feature vector in the dataset to obtain a cluster with a different distance from the query image, and select a negative sample from the low correlation cluster different from the query image. 

The cluster obtained by calculating the Euclidean distance is shown in Figure 5, where panel q is the query image, and panels a–f are negative clusters that observe far distance to the query image. If we want to select five negative examples, then we may first consider panel a, if it is not in the query cluster *q*, or in the positive example clusters, then panel a is used as a low correlation image listed in the input set of query image q. Similarly, panel b is also marked with a low correlation in the input set of images; for image c, although there exists large distance between its feature vector and the feature vector of the query image, panels c and b belong to one labeled cluster. So, panel c, as a low correlation one, is not taken in the negative image set of the query panel q. Panels d–f are taken as low correlation images to *q*. When the number of the required image equals to *N*, the low correlation image is no longer selected, so other images will be no longer considered. If we define *N* as 5, then the selected pictures will be labeled as *N1, N2, N3, N4,* and *N5*. The ranking of *N1* to *N5* is based on Euclidean distance between the hardest sample in each class and the query image.

Entropy weight based on structural distribution: for each query sample *q*, there are a large number of negative samples with different structural distributions. In order to fully utilize them, we propose to weigh the negative samples according to their spatial distribution, i.e., self-similarity and related similarity, that is, the extent to which the sample violates the constraint. The principle of the weighting strategy is shown in Figure 6. The query sample and the positive sample are represented by red circles, *N1* to *N5* represent negative samples, and the left graph shows that when *N2* to *N5* negative samples are far away from the query sample, the *N1* distance query sample when the distance is relatively close, in which *N1* is the hardest negative sample. At this time, keep the distance between *N1* and the query sample, and draw other negative samples. As shown in the right figure, it is found that *N1* is not the hardest negative sample, that is, relatively similar. The decrease in sex shows that the sample distribution around the negative sample has a huge impact on the similarity evaluation of the negative sample and the positive sample. In order to be able to highlight this effect in the process of training, we propose an entropy weight based on structural distribution, the formula of which is shown in Equation (9)
(9)$$\omega =\frac{e^{\beta \left(S_{qi}-\lambda \right)}}{1+\sum_{i\in N_{i}}e^{\beta }\left(S_{qi}-\lambda \right)},$$
where *q* is the sample picture of the query, *i* is the selected sample, and *q* and *i* selected here are negative sample pairs,$\;N_{i}$ is a negative sample set, we define the similarity of the two samples as $S_{\mathrm{qi}}:=\;<f\left(x_{q};\theta \right),\;f\left(x_{i};\theta \right)\;>$, where 〈·,·〉, resulting in an n×n similarity matrix S whose element at (q, i) is $S_{qi}$, where *λ*, *α*, *β* are fixed hyper-parameters. The harder the negative sample is, the larger $S_{qi}$ is, the ω is ranking from large to small, and the serial number is the value of *α*.

#### 3.2.1. The Process of Distributed Structure Learning Loss

Given a query $X_{i}$, we rank them according to their similarity with the query sample and the structural distribution, as illustrated in Figure 7. In each set of pictures, there is one positive sample and five negative samples, and the query sample is represented by *Q*, *P* represents a positive sample, and *N1*, *N2*, *N3*, *N4,* and *N5* respectively represent negative samples with different similarities to the query picture, where *N1* indicates the highest degree of similarity with the query picture, and the function of the loss function is defined as follows. First, the positive sample *P* is pushed to the query sample *Q*, and the distance from the query sample *Q* is within *β*, instead of infinitely close, satisfying the first constraint, then *N1* should be closer to the query picture *Q* than *N2*, push *N1* closer, implement the second constraint on the other images, and so on.

Proper sampling of the sample speeds up the convergence of the model and improves the performance of the model, in this paper, we create a tuple dataset ($X_{q},X_{i}$), where *q* is the query image, and *i* represents an image that matches the query image. All these tuples formed image pairs for training.

#### 3.2.2. The Training Process of DSLL

In this section, we describe and illustrate the training process by using our proposed DSLL, as shown in the algorithm presented below. First, we performed feature extraction on all samples, and used the extracted feature vector to calculate the Euclidean distance between the query sample and the samples in the dataset and ranking online. Then, we selected positive and negative samples according to the requirements mentioned in Section 3.2.1, and perform feature extraction on the selected samples. Finally, the extracted features were used to calculate the loss and the training network is optimized by loss.

The detailed action process for DSLL is as follows:

First, we set the parameters: the number of negative sample *n*, the sorting number of negative sample *a*, learning rate *η*, the initial weights *δ*, the structure distribution entropy weight *ω*, the initial biases *b*, then input *q* (query sample), *i* (retrieve image), the learning rate *η*, the embedding $x_{i}$

Sample pair selection: 

(a) Find the set of samples *S* based on the prior knowledge, such that $\bar{x_{i}}$ is deemed similar to $\bar{x_{q}}$.

(b) Make the sample $\bar{x_{q}}$
in pair with the rest training samples and label these pairs so that: $y_{\left(q,i\right)}=1$ if *Xi* belongs to *S*, and $y_{\left(q,i\right)}=0$ otherwise.

Form the labeled training set by combining all the pairs. 

Training process:

Step 1: feedforward all images into f to obtain the images’ embedding $x_{i}$

Step 2: calculate the distribution of samples around each negative sample, i.e., the structural distribution entropy weight ω, online iterative ranking, and get the images’ number of sorting (a)repeat until convergence:

(a) For each pair ($\bar{x_{q}}$, $\bar{x_{i}}$) in the training set, do

If $y_{\left(q,i\right)}=1$, then update *δ* to decrease
(10)$$L_{\left(\delta ,b\right)}=\frac{1}{2}||\bar{x_{q}}-\bar{x_{i}}-\beta ||{}^{2}{}_{+}$$

If $y_{\left(q,i\right)}=0$, then update δ to increase
(11)$$L_{\left(\delta ,b\right)}=\frac{1}{2}{\left(max\left\{e^{\frac{a}{n}}\tau -||\bar{x_{q}}-\bar{x_{i}}||\right\}\right)}^{2},$$

Step 3: gradient computation and back-propagation to update the parameters of *δ*, *b*.
(12)$$\delta^{\prime }=\delta -\eta \frac{\partial_{L_{\left(\delta ,b\right)}}}{\partial_{\delta }},$$
(13)$$b^{\prime }=b-\eta \frac{\partial_{L_{\left(\delta ,b\right)}}}{\partial_{b}},$$

Step 4: output: updated *L*.

## 4. Experiments

This section aims to evaluate our proposed DSLL model and make a comparison to the current model of state-of-the-art. We first introduced the experimental setup which included the datasets and the protocols of evaluation. Then we analyzed the parameters proposed in the method. In addition, we combined different aggregation methods (including SPoC, MAC and GeM) and whitening methods (PCAw and Lw) with our model to select the method that is most beneficial to our model. At the end of this section, a comparison with existing methods and a visual representation of the results will be presented.

### 4.1. Training Datasets

We adopted the training dataset constructed by Schonberger et al. [37]. The dataset includes 7.4 million images, which are searched and downloaded by Flickr which includes popular landmarks such as cities and countries worldwide. We reconstructed the 3D model by using the dataset with BoW and structure-from-motion (SfM), and used a method exempt from manual annotation to automatically obtain a large dataset with a query image, a positive image, and a cluster with serial 246 number. 

There were a total of 91,642 training images in the dataset, and 98 cluster images identical or nearly identical to the test dataset can be excluded through image retrieval based on BoW. About 20,000 images were selected as query images, 18,1697 pairs of positive images, and 551 training clusters, including more than 163,000 from original datasets, by the minimum hash and spatial verification methods mentioned in the clustering procedure [51]. The original datasets contained all the images of the Oxford 5k and Paris 6k datasets.

### 4.2. Training Configurations

In the experiments, our proposed method applies to all fully convolutional CNN [52]. To train the proposed model, we used the pytorch deep learning architecture to train this deep network model based on DSLL. We initialized the parameters of the networks by the weights of the corresponding network which were pretrained on ImageNet [53].

The first-moment estimate and the second moment estimate of the gradient are dynamically adjusted for the learning rate of each parameter. Since the network pre-training parameters [39] were used, the learning rate was equal to lr = 10^–6^, for the AlexNet network during training. The learning rate was equal to lr = 7 × 10^–7^ for the VGG16 and Resnet101 networks, and 7, with momentum 0.9, justified by the increase in the dimensionality of the embedding. All the images in the training set have been readjusted to a maximum size of 362 × 362 under the premise of ensuring the original aspect ratio. The training results take the experimental data obtained during the 30 epochs.

The experimental environment was an intel(R) i7-8700 processor, GPU with 11GB of memory, NVIDIA(R) 2080Ti graphics card, driver version 419.**, operating system Ubuntu 18.04 LTS, pytorch version v1.0.0, CUDA version 10.0, cudnn version 7.5. 

### 4.3. Test Datasets

To evaluate the impact of our proposed metric learning-based distribution loss on image retrieval in an instance-level image, we used three publicly available datasets, including the dataset of Oxford 5k building dataset [1], the Paris 6k dataset [24], and the Holiday 1k dataset [25]. 

The Oxford 5k building dataset is a widely used landmark dataset consisting of 5062 building images, collected from the Flickr dataset, corresponding to 11 famous landmarks in the Oxford area, with 55 query images evenly distributed across the 11 landmarks.

The Paris 6k dataset consists of 6392 Paris travel images, including Eiffel Tower, etc. Similar to the Oxford 5k dataset, there are also 55 search images.

The Holiday 1k dataset is a landscape-based holiday photo with a total of 1499 images, including 500 magazines and corresponding 991 related images.

We combine these datasets with 100k distractors from Oxford 100k for larger-scale evaluation.

### 4.4. Performance Evaluation Metrics

Throughout the experiment, we used Euclidean distance to measure similarity, and use the standards provided on the dataset website, namely calculating mean average precision (mAP) of the search results. The mAP value is calculated as shown in equation (14): Where AP (*q*) is mAP of the results of the query image compared with the benchmark annotations in the dataset.
(14)$$\mathrm{MAP}=\bar{P}\left(r\right)=\sum_{i=1}^{N_{q}}\frac{P_{i}\left(r\right)}{N_{q}},$$
where $\bar{P}\left(r\right)$ refers to the average precision when the full rate is *r*, $P_{i}\left(r\right)$ refers to the precision of the *i*-th query when the full rate is *r*, $N_{q}$ is the number of queries.

### 4.5. Results and Analysis

#### 4.5.1. The Impact of Margin Parameter τ

As shown in Section 3, for each query, DSLL can guarantee the structural similarity order consistency of negative samples. Specifically, we ensure the consistency of sample structure similarity by adjusting the size of the negative sample structure space. Since the size of the negative sample space is determined by the constraint parameter τ, as shown in Figure 8, we conducted experiments on the large dataset Oxford 5k and Paris 6k to analyze their impact.

In order to find the threshold τ suitable for different data sets and can improve the performance of the network, we select the value from 0.5–2.0 to experiment, and select the results when τ = 0.75, 1.00, 1.25, 1.50 according to the experimental results. The data shown in Figure 8a,b are trained under AlexNet, while Figure 8c,d represent the data obtained under VGG training, and the results show that the performance is the best when τ = 1.25. It can be seen from the figure that the performance of the network increases with the increase of the threshold τ, but when the τ increases to 1.45, the map value decreases, because the larger threshold τ pulls the negative samples with high similarity away, affecting the training effect. In the following, we choose the experiment when the threshold τ is 1.25.

#### 4.5.2. The Role of Structural Similarity Ranking Consistency

In this experiment, we use the contrast loss function as a prototype to compare the retrieval performance without adding ranking with Euclidean distance and structural similarity measure. The search performance results are shown in Figure 9. From the figure, we can see that the result of adding the ranking loss function is better than the original model. Among them, adding the ranking loss measured by structural similarity is the best. The reason why it is better than the ranking of Euclidean distance is that it considers the spatial distribution of the feature and the spatial distribution of the samples around it, which can ensure the vector structure of the negative sample to the greatest extent. However, the ranking of the Euclidean distance dimension only considers the Euclidean distance, and the idle features cannot be well characterized.

#### 4.5.3. Imapact of Margin Parameter β

To study the influence of the threshold parameter β, we set t = 1.5 in the experiment. The results are presented in Figure 10. We observe that a proper positive constraint β is important for *DSLL* to learn discriminative embeddings. Since β controls the extent to which this is being approximated, in later experiments, we chose β = 0.15 for the experiment.

#### 4.5.4. The Combined Impact of DSLL

To verify the effect of our newly proposed loss function based on structural consistency and similarity ranking consistency, we compare the original contrast loss, the loss function with structural similarity ranking consistency, and the retrieval performance of the loss function with structural consistency and structural similarity ranking consistency. The experimental results are shown in Figure 11, where None refers to the original contrast loss, A represents the loss function based on structure preservation, and B represents the loss function based on structural similarity ranking consistency.

#### 4.5.5. Comparison of MAC, SPoC and GeM

This section compares state-of-the-art representations: MAC, SPoC and GeM. We use the *DSLL* function to perform end-to-end network training on the data set. The pooled method of comparison is connected to the last layer of the AlexNet network convolutional layer. It can be concluded from Figure 12 that GeM is superior to MAC and SPoC on all data sets.

#### 4.5.6. Comparison of PCA-Whitening and Learned Discriminative Whitening

The whitening proves to be essential in some cases of CNN-based descriptors. In this experiment, we chose the best whitening method for our network. Also, we compare the existing PCA-whitening [32] (PACw) with learned discriminative whitening [37] (Lw) and the results without post-processing. Figure 13 shows the results of the comparison. Our experiments show that PCAw usually degrades performance, while Lw achieves the best performance in most cases and never achieves the worst performance. Therefore, we choose Lw because it has the best performance and the best generalization ability.

#### 4.5.7. Gradient Value Selection

This section compares the performance of two pooling method including MAC pooling and GeM pooling in combination with PCAw and Lw in different dimensions. The performance for varying descriptor dimensionality is plotted in Figure 14, As can be seen from the figure, the higher the dimension, the better the performance. When the same pooling method is used, the Lw effect is better than PCAw, and the performance is best when combined with GeM pooling. In general, when the dimension is 256, the combination of GeM pool and Lw has the best performance.

#### 4.5.8. Comparison with the State of the Art

This section compares the performance of our proposed DSLL method with the updated representations of the state-of-the-art performance. Table 1 lists the performance comparisons. We divide the network into two categories, (1) the use of the fine-tuned network (yes), and (2) the use of the off-the-shelf network (no), so that we can observe that our proposed DSLL was superior to all previous prior art methods. When using the VGG network framework, compared with the RMAC [33], DSLL provides a significant improvement of +5.3%, +1.8% and +1.7% in mAP on Oxford 5K, Paris 6k, and Holiday 1k datasets. Furthermore, the DSLL signatures achieves a gain of +0.5%, +1.2% and +1.3% on Oxford 5K, Paris 6k and Holiday 1k datasets, which surpassed recently published [5]. When using the ResNet network framework, on Oxford 5K, Paris 6k, and Holiday 1k datasets, the experimental results achieved +0.6%, +1.2% and +0.4% growth compared to GeM. The improvement in retrieval accuracy, between and GeM, is even more significant on large scale datasets: Holidays 101k (+3.2%) under the VGG and Holiday 101k (+2.8%) under the ResNet framework. Re-ranking and query extensions [13,37] have recently become standard techniques for improving searchability. We applied QE to the *DSLL* representation, which can be can be observed from Table 1 that our method outperforms state-of-the-art results. Under the VGG framework, the gain over GeM+αQE [37] is +2.9% and +1.9% on the Paris 6k dataset. Compared to R-MAC+QE [33] the gain was 3.1%, 2.6%, 3.6%, and 9% on Oxford 5k, Oxford 105k, Paris 6k, and Paris 106k datasets. Under the ResNet framework, the DSLL architecture achieved mAP of 92.4%, 90.0%, 96.7%, 93.9% and offered over 91.0%, 89.5%, 96.7%, 91.9% gain over the GeM+αQE [37] on Oxford 5k, Oxford 105k, Paris 6k and Paris 106k datasets.

#### 4.5.9. Visualization Purposes

In order to visualize the search results, as shown in Figure 15, we present quantitative results based on several query sample. These demonstrate that retrieved images from the top five were similar to the query image by turning to features extracted by fine-tuned, pretrained and DSLL-based ResNet101 networks respectively. In Figure 15, the first line is the result on Oxford datasets, and the query image is from Hertford Bridge, Oxford; the results of Paris datasets are displayed in the middle column, the query image is from Arc de Triomphe, Notre Dame de Paris; the last column Is the result of the query of Holiday datasets, and query image is from the field and snowy mountains.

DSLL-based features significantly outperform those pretrained and fine-tuned features. The results indicate that DSLL algorithm based on structural consistency and structural similarity ranking can distinguish images with high within class variance by well learning high-level semantic features.

## 5. Conclusions

In this paper, we proposed a novel DSLL algorithm based on structural preservation and structural distribution consistency. It learns the hierarchical structure that represents the deep features of image samples in two ways. First, it preserves the structural information of the positive samples by learning the hyperplane for each query sample in the model, and then the entropy weight based on structural distribution calculates the spatial relationship between the negative samples and the surrounding samples to obtain the weights. The eigenvectors are combined with the weights to train the network, and the accuracy of the retrieval is improved by realizing the structural preservation of the image feature vectors and the consistency of the structural similarity ranking. We train the whole framework in an end-to-end fashion, and the experimental results from broad tests prove that DSLL achieves the most state-of-the-art performance.

## Figures and Tables

**Figure 1 entropy-21-01121-f001:**
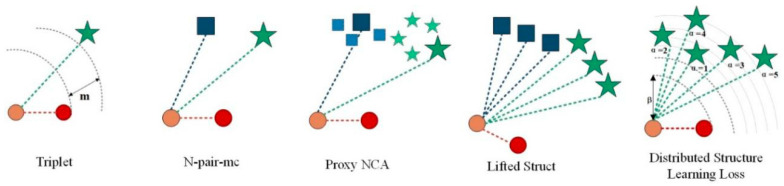
A graphical representation of the loss of different metric learning is presented. Different categories are represented by different shapes. For the sake of simplicity, we only represent three classes. Anchor (query picture) is represented in brown. In the triplet loss [7], anchor is only compared to one positive sample and one negative sample. In N-pair-mc [20], proxy-NCA [10], and lifted struct [22], a positive sample and multiple negative sample types were introduced. N-pair-mc randomly selects a negative sample in each negative sample class. The proxy NCA pushes the negative sample set agent away from the anchor instead of pushing the negative sample farther away. Lifted struct uses all negative sample categories. Instead, our proposed distribution structure learning loss uses a representative positive sample and five negative samples with different similarities to the anchor.

**Figure 2 entropy-21-01121-f002:**
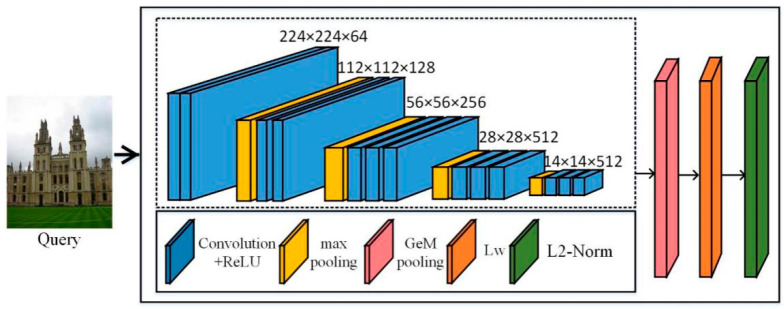
Built convolutional neural network (CNN) network architecture.

**Figure 3 entropy-21-01121-f003:**
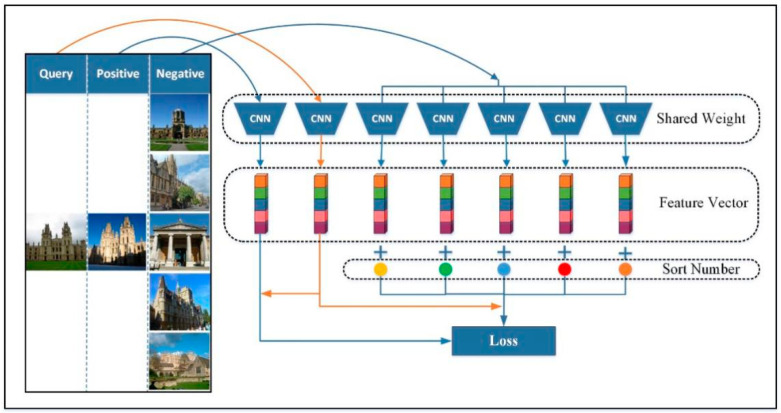
Network training using distribution structure learning loss (DSLL) loss.

**Figure 4 entropy-21-01121-f004:**
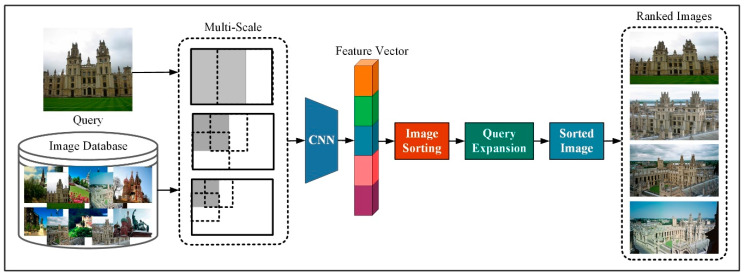
Evaluation of DSLL using mean average precision (mAP).

**Figure 5 entropy-21-01121-f005:**
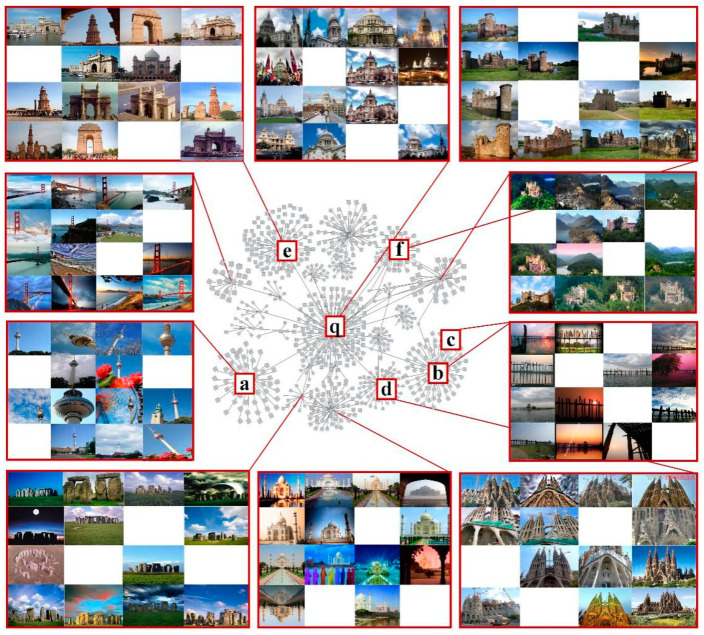
The selection of negative images: every cluster is formed by similar images; (**a**–**f**) are the negative image with low similarity with the query image; (**q**) is the query image.

**Figure 6 entropy-21-01121-f006:**
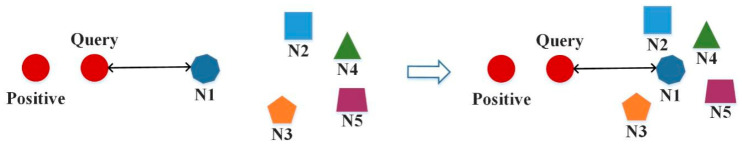
The effect of the distribution of nearby samples on the sample: the relative similarity of the negative samples decreases as the similarity of other negative samples increase.

**Figure 7 entropy-21-01121-f007:**
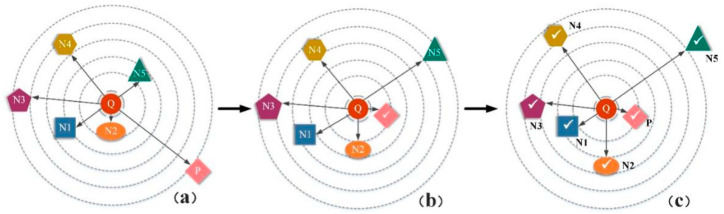
Process of DSLL function of a synthetic batch. (**a**) Initial sample distribution before learning. (**b**) The distribution of results by pulling the positive sample closer within the threshold boundary. (**c**) The final results making negative samples maintain structural similarity. The overall embedding by enforcing Equation (7). Symbol √ indicates completion of learning.

**Figure 8 entropy-21-01121-f008:**
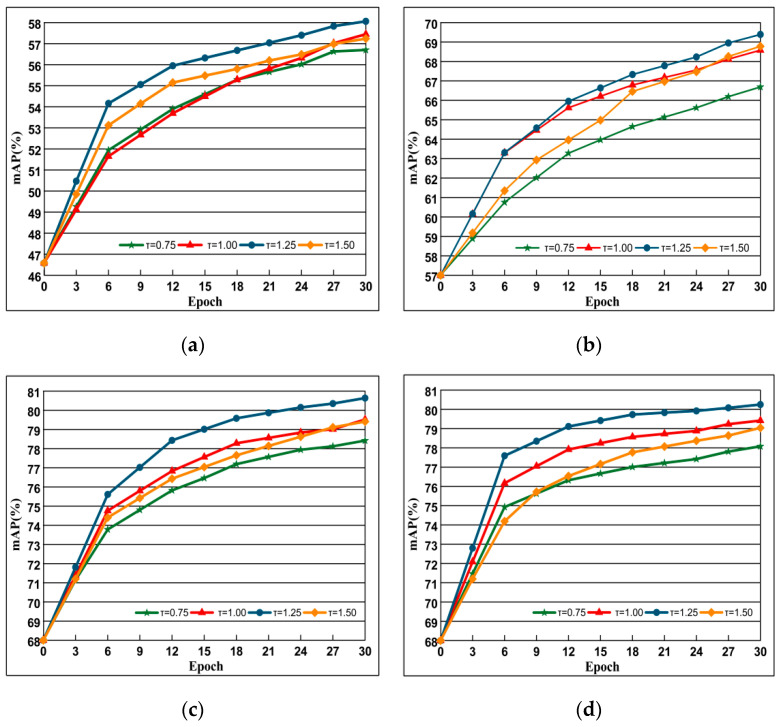
The impact of choice of the different τ selection. Evaluation is performed with AlexNet (**top**) and VGG (**bottom**) on Oxford 5k and Paris 6k datasets. The curve line presents the evolution of mAP depending on training epochs. Epoch reflects off-the-shelf network. (**a**) Oxford 5k; (**b**) Paris 6k; (**c**) Oxford 5k; (**d**) Paris 6k.

**Figure 9 entropy-21-01121-f009:**
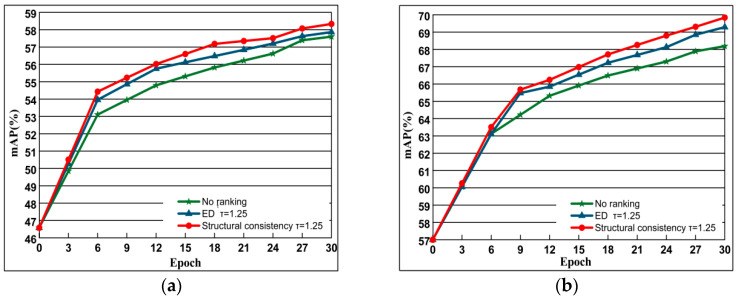
The comparison of performance based on methods of no ranking, ED and Structural consistency. where no ranking represents the unbalanced contrast loss function, ED represents the addition of a loss function with a measure of Euclidean distance, and Structural consistency indicates the addition of structural similarity. Evaluation is based on the performance with AlexNet of datasets Oxford 5k and Paris 6k. The curve line presents the evolution of mAP depending on training epochs. (**a**) Oxford 5k; (**b**) Paris 6k.

**Figure 10 entropy-21-01121-f010:**
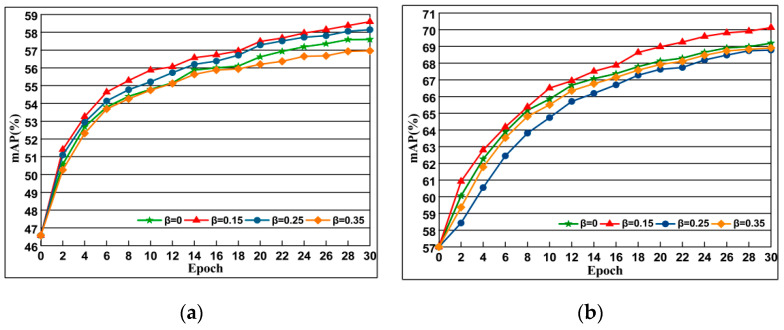
The impact of choice of the different β selection of which the evaluation is performed along with AlexNet on Oxford 5k and Paris 6k datasets. The curve line presents the evolution of mAP depending on training epochs. Epoch reflects off-the-shelf network. (**a**) Oxford 5k; (**b**) Paris 6k.

**Figure 11 entropy-21-01121-f011:**
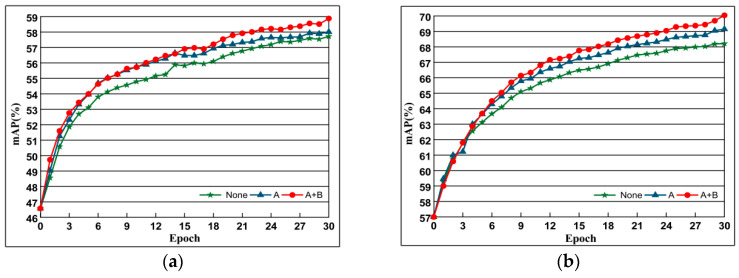
The comparison of performance based on methods of none, A and A + B. Evaluation is based on the performance with AlexNet of datasets Oxford 5k and Paris 6k. The curve line presents the evolution of mAP depending on training epochs. (**a**) Oxford 5k; (**b**) Paris 6k.

**Figure 12 entropy-21-01121-f012:**
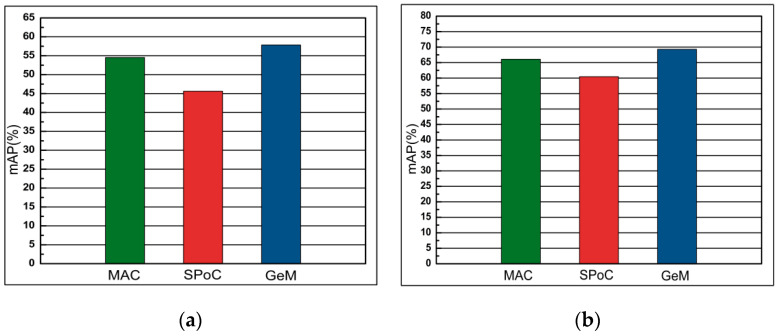
The comparison of performance (mAP) of different pooling layers fine-tuned with AlexNet of datasets Oxford 5k and Paris 6k. (**a**) Oxford 5k; (**b**) Paris 6k.

**Figure 13 entropy-21-01121-f013:**
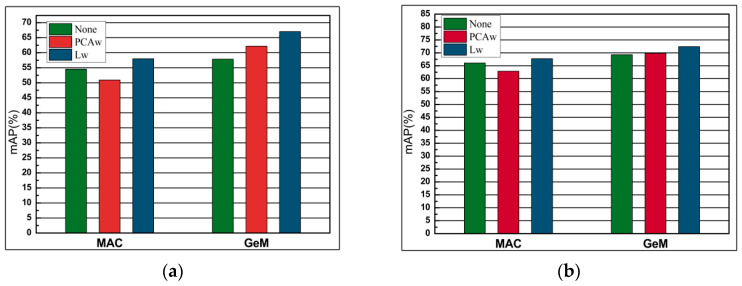
The comparison of performance (mAP) of post-processing of CNN vector, without post-processing, principal component analysis (PCA)-whitening (PCAw) and learned whitening (Lw). The performance of evaluation is with AlexNet on Oxford 5k and Paris 6k datasets. (**a**) Oxford 5k; (**b**) Paris 6k.

**Figure 14 entropy-21-01121-f014:**
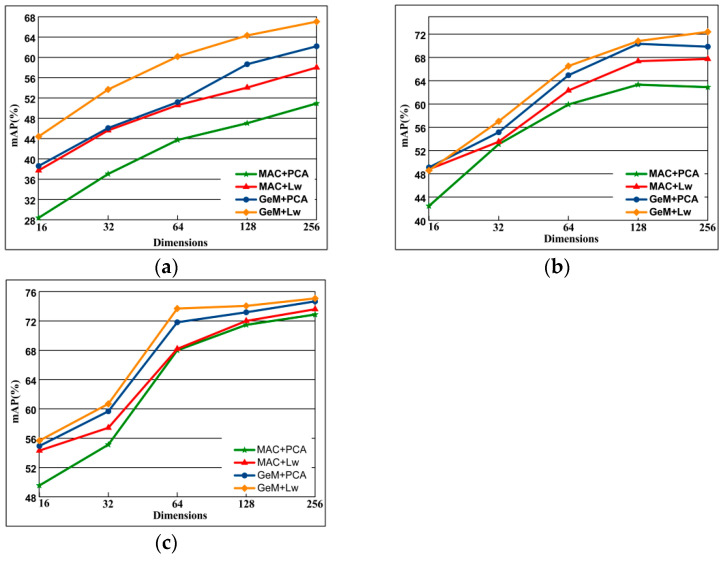
The comparison of performance on dimensionality reduction is performed by PCAw and Lw with the fine-tuned Alexnet with MAC layer and GeM layer on Oxford 5k, Paris 6k, and Holiday 1k datasets. (**a**) Oxford 5k; (**b**) Paris 6k; (**c**) Holiday 1k.

**Figure 15 entropy-21-01121-f015:**
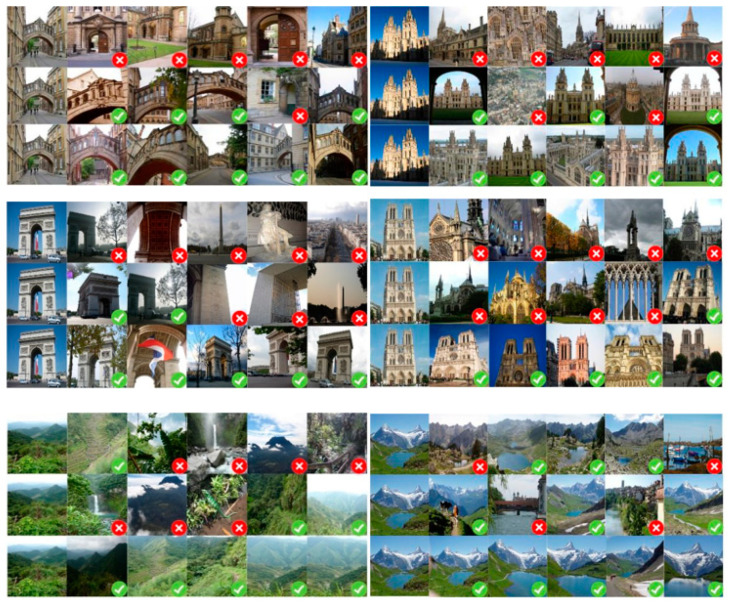
Top five retrieval results for the Oxford (the first line), Paris (the second line) and Holiday (the third line) datasets. For both columns, the first image is the query image, and for each query, the three rows represent results of the extracted features from pretrained, fine-tuned, and DSLL-based ResNet101 respectively. The green ticks on the right bottom of the picture refer to the correct results, while the red ticks mean false results.

**Table 1 entropy-21-01121-t001:** Comparison of performance (mAP) between our method and the state-of-the-art image retrieval method under VGG and ResNet (Res) deep network. The best result is shown in red.

Net	Method	F-Tuned	Oxford5k	Oxford105k	Paris6k	Paris106k	Holidays	Hol101k
**Compact Representation using Deep Networks**								
VGG	MAC [54] †	no	56.4	47.8	72.3	58.0	79.0	66.1
	SPoC [30] †	no	68.1	61.1	78.2	68.4	83.9	75.1
	CroW [55]	no	70.8	65.3	79.7	72.2	85.1	-
	R-MAC [24]	no	66.9	61.6	83.0	75.7	86.9	-
	BoW-CNN [28]	no	73.9	59.3	82.0	64.8	-	-
	NetVLAD [38]	yes	71.6	-	79.7	-	87.5	-
	Fisher Vector [56]	yes	81.5	76.6	82.4	-	-	-
	R-MAC [33]	yes	83.1	78.6	87.1	79.7	89.1	-
	GeM [37]	yes	87.9	83.3	87.7	81.3	89.5	79.9
	*ours	yes	88.4	84.7	88.9	81.5	90.8	83.1
Res	R-MAC [24] ‡	no	69.4	63.7	85.2	77.8	91.3	-
	GeM [37]	yes	87.8	84.6	92.7	86.9	93.3	87.9
	*ours	yes	88.4	84.9	93.9	87.8	93.7	90.7
Re-ranking (R) and Query Expansion (QE)								
VGG	CroW+QE [55]	no	74.9	70.6	84.8	71.0	-	-
	R-MAC+R+QE [24]	no	77.3	73.2	86.5	79.8	-	-
	BoW-CNN+R+QE [28]	no	78.8	65.1	84.8	64.1	-	-
	R-MAC+QE [33]	yes	89.1	87.3	91.2	86.8	-	-
	GeM+αQE [37]	yes	91.9	89.6	91.9	87.6	-	-
	*ours	yes	92.2	89.8	94.8	89.5	-	-
Res	R-MAC+QE [24] ‡	no	78.9	75.5	89.7	85.3	-	-
	R-MAC+QE [13]	yes	90.6	89.4	96.0	93.2	-	-
	GeM+αQE [37]	yes	91.0	89.5	95.5	91.9	-	-
	*ours	yes	92.4	90.0	96.7	93.9	-	-

†: The results of our evaluation of average pooling (SPoC) and mac using PCAw and off-the-shelf networks. ‡: Results of evaluating R-MAC using [13] and off-the-shelf networks.

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
