# Peer review of "Distribution Structure Learning Loss (DSLL) Based on Deep Metric Learning for Image Retrieval"

_entropy, 2019, doi:10.3390/e21111121_

Round 1
Reviewer 1 Report
This paper introduces a distribution structure learning loss (DSLL) for deep metric learning methods, where DSLL aims to preserve the geometric information of images. Experimental results on three building datasets demonstrate the performance of the DSLL. The paper is somewaht novelty, and the experimental evaluation is sufficient. The main concerns of this paper are summarized as: 1) There are some typos should be checked and corrected, e.g., line 61:"for known query image” --> "for a known query image"; line 94: "Section II" --> "Section 2", "Section III" --> "Section ", and others. 2) Please check whether the formulation of Lifted structured loss (Eqs. (4) & (5)) is correct or not. 3) In Eq. (7), should [d_{(q,i)} - \beta] be [d_{(q,i)} - \beta]_{+}? the parameter \tau does not appear. 4) In Figure 5, more details should be added to figure captation. 5) In Section 3.2.2, how to calculate similarity and get N1, N2, ..., N5? 6) There are some works on deep metric learning can be cited in the paper, e.g., Duan et al., Deep Localized Metric Learning, TCSVT, 2018; Hu et al., Sharable and Individual Multi-View Metric Learning, TPAMI, 2018; Lu et al., Discriminative Deep Metric Learning for Face and Kinship Verification, TIP, 2017.
Author Response
Dear Reviewer:
We are very grateful for your meaningful and in-depth comments. With your tips and suggestions, we have rethought the article, enriched the content, and improved the deficiencies and corrected some errors. For your specific questions, we have answered in detail below.Please see the attachement

Reviewer 2 Report
This paper presents a new deep metric learning algorithm for image retrieval. The authors introduce the concept of structural preservation and structural consistency. An entropy weight based on structural distribution is also incorporated to the loss function. Experimental results on benchmark datasets are reported and discussed.
Pros.
The idea is well motivated and the paper is clearly organized. Section 2 presents a nice review of deep metric learning algorithms. Promising results are observed on multiple datasets. Analysis of parameter sensitivity is also provided.Cons.
The paper presentation shall be improved. For example, the steps in Section 3.2.3 shall be summarized in an Algorithm. Also, the format of many references are incorrect. The loss function of the proposed method should be clearly presented. The title of Section 4 should be Experiments. More related work on metric learning or deep metric learning could be added, such as [a-d].[a] Stochastic Class-Based Hard Example Mining for Deep Metric Learning. CVPR 2019.
[b] Retrieving and Classifying Affective Images via Deep Metric Learning. AAAI 2018.
[c] SLMOML: Online Metric Learning with Global Convergence, IEEE Trans. Circuits and Systems for Video Technology (T-CSVT), 18(10): 2460-2472, 2018.
[d] Scalable Large Margin Online Metric Learning, International Joint Conference on Neural Networks (IJCNN), 2016.
Author Response
Dear Reviewer:
We are very grateful for your meaningful and in-depth comments. With your tips and suggestions, we have rethought the article, enriched the content, and improved the deficiencies and corrected some errors. For your specific questions, we have answered in detail below. Please see the attachment.
